# Population Decline through Tourism Gentrification Caused by Accommodation in Kyoto City

**Haruka Kato ***  **and Atsushi Takizawa**

Department of Housing and Environmental Design, Graduate School of Human Life and Ecology, Osaka Metropolitan University, Osaka 5588585, Japan
* Correspondence: haruka-kato@omu.ac.jp; Tel.: +81-6-6605-2823

**Abstract:** Tourism gentrification has become a social issue in tourist cities worldwide. This paper's research question is as follows: has tourism gentrification caused a population decline in tourist cities? This study aims to clarify the statistical relationship between the population decline and the location of accommodation on the neighborhood association scale. It analyzes Kyoto City, which is one of the most famous tourist cities worldwide. The statistical relationship between two types of accommodation—hotels and simple accommodation—is analyzed, using geographic natural experiments. The study concludes that the neighborhood association with simple accommodation decreased the population more significantly than that without simple accommodation in the historical center of Kyoto City. This result indicates that the tourism gentrification had caused a population decline in the historical center of Kyoto City. Moreover, it was found that tourism gentrification has affected the outside center of Kyoto City. The population decline might be due to simple accommodation being converted from houses due to tourism gentrification. This study's results suggest the need for urban policy to regulate zoning for the locations of simple accommodation.

**Keywords:** tourism gentrification; center of Kyoto City; simple accommodation; population decline

## 1. Introduction

### 1.1. Background

Tourism gentrification has become a social issue in tourist cities worldwide. Tourism gentrification is one type of gentrification, which is the reinvestment of capital in the urban center to produce space for a more affluent class of people than that which currently occupies that space [1]. Regarding the reinvestment of capital, touristification has generated a positive economic cycle from abroad capital, which might revalorize real estate and appropriate urban surplus [2]. However, regarding recent touristification, new forms of tourism using digital platforms might cause the risk of tourism gentrification in some neighborhoods of tourist cities [3]. Gotham [4] defined tourism gentrification as a heuristic device to explain the transformation of a middle-class neighborhood into a relatively affluent and exclusive enclave marked by a proliferation of corporate entertainment and tourism venues. As a specific feature of tourism gentrification in recent years, Stors and Kagermeier [5] pointed out that sharing technologies such as Airbnb have caused neighborhood changes. In other words, the conversion of homes to accommodation has led to reinvestment and the transformation of neighborhoods into tourist destinations [6].

This paper's research question is as follows: has tourism gentrification caused a population decline in tourist cities? Our hypothesis is that there is a statistical relationship between population change and the location of accommodation on a neighborhood scale. It was reported that gentrification caused the displacement of residents who lived in a neighborhood long term [7]. That is because gentrification entails middle-class settlement in renovated or redeveloped properties in older, inner-city districts formerly occupied by a lower-income population [8]. Therefore, tourism gentrification also might cause the

displacement of long-term residents, resulting in the conversion from houses to accommodation. However, it was also reported that large numbers of households are unaffected by displacement [9]. The discussion on gentrification has sometimes become emotional because of the distress it causes [10].

Therefore, the theoretical framework of this study is focused on population decline due to tourism gentrification though the displacement of residents. That is because the driver of recent tourism gentrification is not simply an outcome of a globalized 'rent gap' by the middle-class settlement, but results from new forms of tourism and short-term rentals by digital platforms [3]. Population decline is a significant issue in urban planning. Well-known examples are the urban plannings of the Rust Belt in the United States, due to economic decline [11], those of former East Germany, due to political changes [11], and those of Japan, due to an aging population [12]. In addition, during the COVID-19 pandemic, population decline due to urban exodus has also been focused upon [13]. It was also a significant issue in the urban planning of tourist cities due to tourism gentrification. Regarding tourism gentrification, clarifying whether the population declined locally or across the city is important for urban planning. This study's finding on the statistical relationship between population decline and the location of accommodation could help one to decide whether or not to develop urban policies for preventing population decline due to tourism gentrification.

### 1.2. Purpose

This study aims to clarify the statistical relationship between population decline and the location of accommodation on the neighborhood association (NA) scale. The result would indicate the tourism gentrification causes population decline, which is the research question in this manuscript. The methodology used was geographic natural experiments, which is a method that analyzes causal relationships. In this study, the geographic natural experiment compared the population change in NA, with and without accommodation. For the case study, this study analyzed Kyoto City, which is one of the most famous tourist cities worldwide, as shown in Figure 1. The satellite map in Figure 1 is from the open-source map of Arc GIS PRO, and complies with copyright [14].

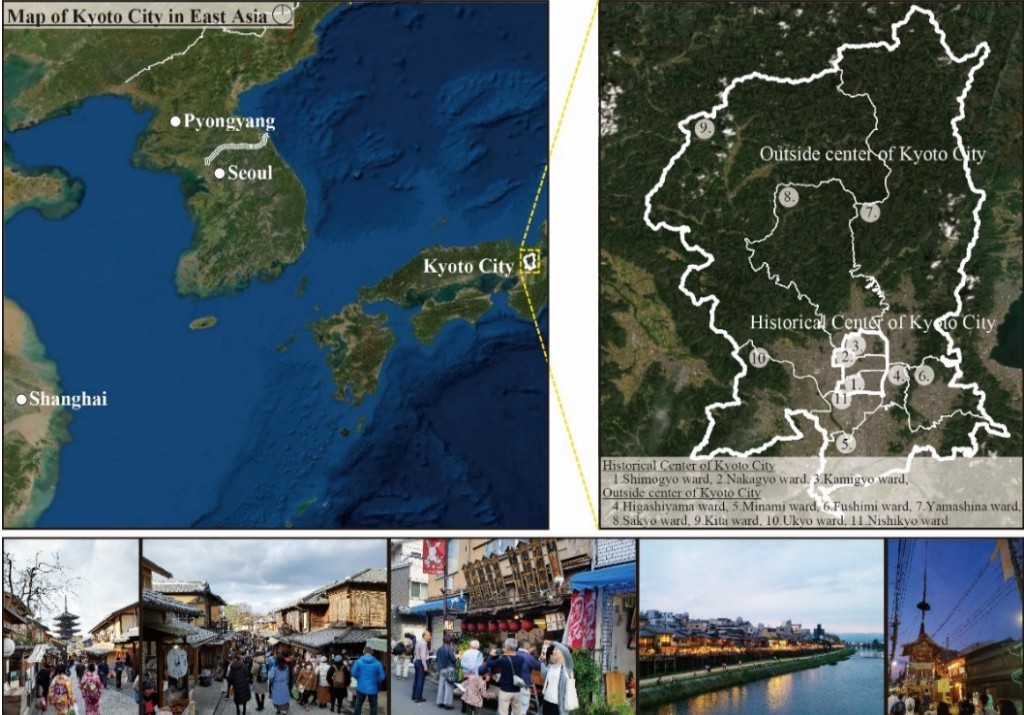

**Figure 1.** Kyoto City in East Asia (Author H.K. took the five photographs).

For the analysis of Kyoto City, this study distinguished between the historical center and the outside center of Kyoto City. The historical center of Kyoto City includes Shimogyo, Nakagyo, and Kamigyo wards. The historical center of Kyoto City has been the capital of Japan for 1200 years [15]. Since early times, people have lived in harmony with nature and have nurtured their culture, such as the Yama-Hoko Events of the Gion Festival [16]. As a representative of history and culture, UNESCO (United Nations Educational, Scientific, and Cultural Organization) has registered the 16 shrines and temples and the one castle in Kyoto City as World Heritage Sites [17]. Therefore, many tourists visit the historical center of Kyoto City.

The analysis period is from April 2015 to March 2020. During this period, the Japanese government took measures to promote tourism in Japan, such as easing tourist visa issuance [18]. Therefore, many tourists visit Kyoto City [19]. On the other hand, excessive touristification has been a social issue for residents. For example, local newspapers have reported that many residents have negatively evaluated this situation, "Is Kyoto City really our town?" [20]. The evaluations suggest that tourism gentrification might be occurring in Kyoto City. However, after March 2020, due to the COVID-19 pandemic, the number of tourists declined rapidly in Kyoto City [19]. Moreover, in Japan, the unit of one year is often set from April to next March. This fact suggests that the analysis period coincided with the period for when tourism gentrification occurred in Kyoto City.

The statistical relationship was analyzed using geographic natural experiments based on the NA scale. This study estimates the causal relationship by comparing the NAs with accommodation and the NAs without accommodation. The reason for the NA scale analysis was related to the community governments. In Kyoto, NAs govern their communities by themselves, such as childcare, welfare for the elderly, traditional events, and disaster prevention [21]. The NA is the foundation upon which communities make decisions about urban planning in Japan [22]. Some studies have analyzed this geographical relationship based on the NA scale [23,24]. Therefore, this study examined tourism gentrification based on the NAs.

This study analyzed the two types of accommodation—simple accommodation (SA) and hotels—in Figure 2. SA is residential-type accommodation that is registered on Airbnb and the like, using a digital platform [25]. There is no appropriate translation of the SA, which is translated form "kannisyukusyo" in Japanese. For example, the Hotel Business Act in Japan has translated it as "common lodging house" [26], and the Kyoto City Government has translated it as "minpaku accommodation" [27]. This study translates it as SA (simple accommodation), following business practice. SA comprises accommodation facilities that are designated under the Hotel Business Act in Japan [26]. Due to the law, it is easier to obtain business licenses for SA than for hotels. However, before touristification, SA was mainly used for the purpose of providing public shelters [28]. In 2015, along with the popularity of Airbnb and other digital platform services, most SA were used for tourism-related purposes [25]. Due to the rapid increase in SA, SA became so problematic that residents took measures for SA on their own [29]. Therefore, this study analyzes the relationship between SA and hotels.

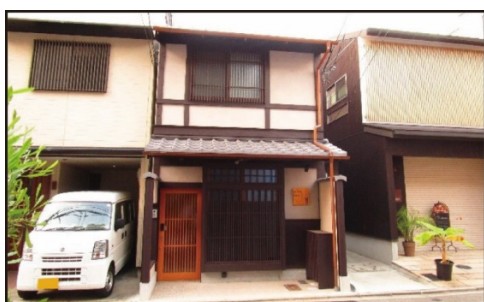 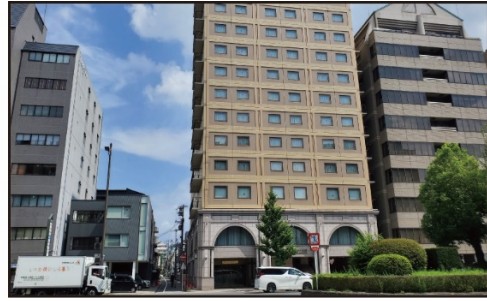

**Figure 2.** Sample of simple accommodation (**left side**) and hotel (**right side**) (Author H.K. photographed the pictures).

### 1.3. Literature Review

Tourism gentrification has been analyzed mainly in Spain, which has tourism gentrification issues. In Palma, González-Pérez et al. [30] clarified the effects of tourism gentrification, such as the financial aspects (real estate speculation and rising property prices), the social aspects (displacements), and the functional and scenic aspects (symbolic and commercial transformation). Regarding the social aspects, Almeida-García et al. [31] clarified that the residents' annoyance caused by tourism gentrification is more intense than tourism phobia. In the center of Madrid, Ardura Urquiaga et al. [32] verified a strong association between the growth in tourist arrivals and the settlement of new residents from wealthy economic backgrounds. Regarding the new residents, López-Gay [33] revealed a process of population restructuring characterized by a decrease in long-term residents and inhabited dwellings, and the arrival of young and transnational gentrifiers that are increasingly mobile and form a transient population, as in the case of the Gòtic area in Barcelona. The results suggest that population decline does not necessarily occur due to tourism gentrification, depending on the cities. However, in the historical center of Seville, Parralejo et al. [34] clarified that population decline was affected by the impact of the growth of tourist accommodation. Unlike [34], this study analyzed the NA of the entire city, including the historical center and the outside center of Kyoto City. Therefore, the novelty of this study lies in the fact that it analyzes the statistical relationship between the population decline and the tourism gentrification caused by simple accommodation in the case of Kyoto City.

Regarding the urban effect of tourism gentrification, in Madrid, Álvarez-Herranz et al. [35] clarified a spillover effect from the central neighborhoods, with a high per capita income, to nearby neighborhoods with a lower income, due to the level of tourist crowding. On the other hand, in Valencia, García-Amaya et al. [36] clarified that simple accommodation is concentrated in specific neighborhoods that are different from the historical center. The result is different from the literature of Álvarez-Herranz et al. [35]. This difference suggests the possibility of regional differences in the impact of tourism gentrification. The novelty of this study is to analyze the statistical relationship by dividing it into the historical central area and the other areas.

### 1.4. Article Structure

This manuscript consists of five chapters based on the IMRAD format: Materials and Methods in Section 2; Results in Section 3; Discussion in Section 4; and Conclusions in Section 5.

## 2. Materials and Methods

### 2.1. Geographic Natural Experiment

The methodology of this study is a geographic natural experiment for the analysis of the research question, whether tourism gentrification causes population declines. The geographic natural experiment is one of analyzing causal relationships between the population decline and the location of the accommodation. Analyzing causal relationships using the randomized controlled trial method is difficult in the field of urban planning. Geographic natural experiments are an effective method for analyzing statistical relationships in urban planning, although they are limited by their inherent locality. The methodology is a powerful tool to clarify causal relationships under conditions that cannot allow for intervention experimentation.

A natural experiment is defined as a credible claim that the assignment of the non-experimental subjects to treatment and control conditions is random, with a focus on the data obtained from naturally occurring phenomena [37]. Natural experiments are used in many studies in fields such as urban political science, which studies the effects of policy interventions [38]. Among some types of natural experiments, this study analyzed geographic natural experiments. Geographic natural experiments investigate geographic or administrative boundaries to construct treated and control groups by exploiting certain

geographical features that generate as-if random variation in the treatment assignment [39]. This study assumed that accommodation is randomly located, and assessed the treated group as NAs with accommodation and the control group as NAs without accommodation. Based on the two groups, this study compared the population change in NAs according to the existence of accommodation.

Regarding the comparison, we conducted the Mann–Whitney U test, which is a nonparametric test to compare outcomes between two independent groups. In this study, the population change from 2015 to 2020 was tested by the groups between NAs with accommodation and NAs without accommodation. The study set the criteria of the *p*-value of the test as 0.01. The NAs, which contain two types of accommodation (SA and hotels), were compared separately regarding the historical center and the outside center of Kyoto City.

### 2.2. Accommodation List

For analyzing the number of changes in accommodation, this study used the 'List of Permitted Facilities Based on the Accommodation Business Law', which was provided by the Kyoto City Information Centre [40]. The Kyoto City Government publishes the available accommodation data that are accepted by applications based on the Ryokan Business Law. In Japan, all accommodation facilities, including those registered with Airbnb, must register with local governments. Therefore, Airbnb accommodation that is not registered with governments is illegal and is rectified by governments. Therefore, the data used in this study covers all accommodations.

The list included attributes such as the type of accommodation (SA/hotels), the location (address), the names of applicants, and the year that the accommodation was turned into a business. The types of accommodation were hotels and SA. The hotels included Japanese traditional style hotels, which are translated as "ryokan" in Japanese. Based on the attribute of the year in which the accommodation was turned into a business, this study extracted the accommodation that started to operate from April 2015 to March 2020. Using the data, this study calculated the number of hotels and SA for each NA. According to this calculation, this study analyzed the existence of hotels and SA for each NA.

### 2.3. Census Data

For analyzing the population change, this study used the Japanese census data [41,42]. The Japanese census has been conducted as a critical national essential statistics survey. Therefore, the Statistics Law in Japan requires all citizens to submit a survey form. The Japanese census is conducted by the government every five years. The data are used for policymaking in Japan. This study analyzed the population for each NA using the census data conducted in 2015 and 2020. The change in the population for each NA was calculated based on the population difference between 2015 and 2020.

## 3. Results

### 3.1. Map of Each Score

Section 3.1 examines the distribution of each score on the map. The accommodation distribution is analyzed in Figures 3 and 4. The satellite maps of Figures 3 and 4 complied with the copyright [14]. Figure 3 shows the distribution of SA. There is a high-density SA throughout the urban areas of the city. It was also found that SA gradually spread from the historical center to the other center of Kyoto City. The number of SA built from April 2015 to March 2020 was 3227. This means that SA increased rapidly from 2015 to 2020. Especially, Figure 3 shows many SA in the historical center and the outside center of Kyoto City.

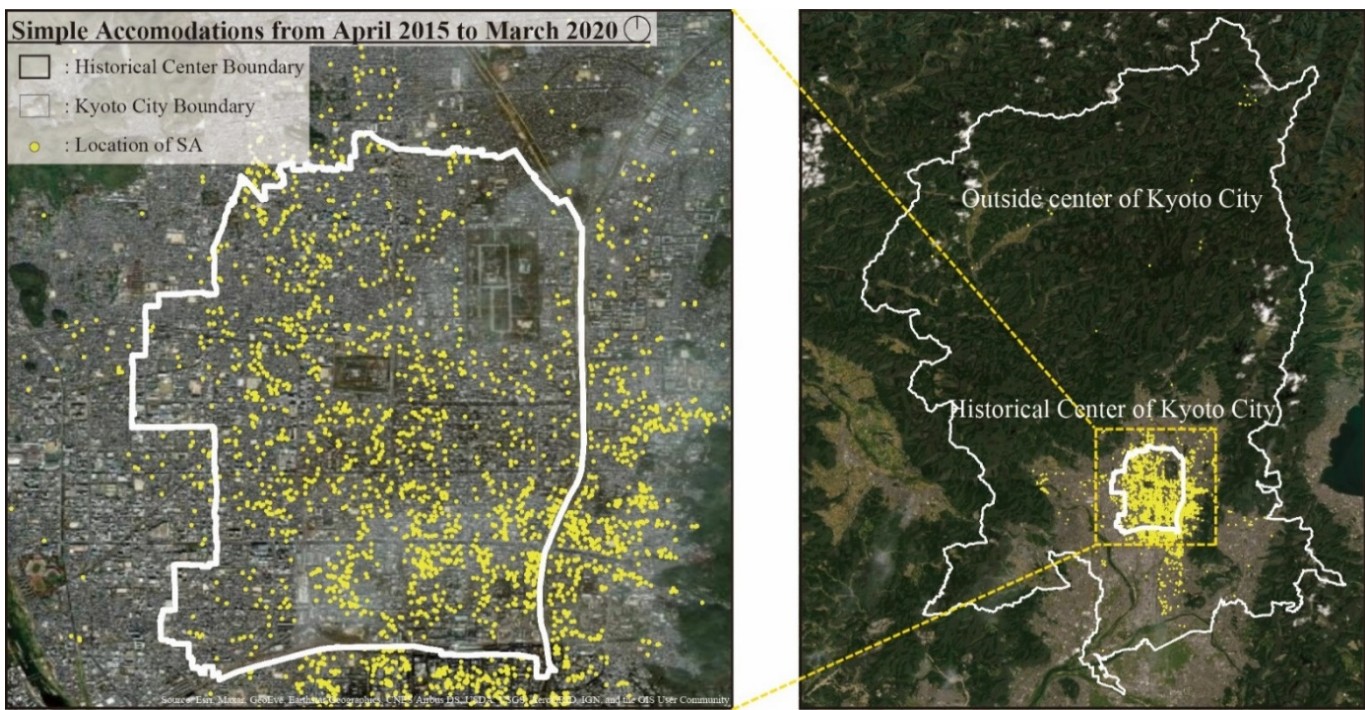

**Figure 3.** Map of simple accommodation in Kyoto City.

Figure 4 shows the distribution of the hotels. Compared to SA, it was found that there are fewer hotels. The number of hotels built from April 2015 to March 2020 is 198. Figure 4 shows many hotels in the outside center of Kyoto City. It was also found that hotels are located in some clusters in the outside center of Kyoto City.

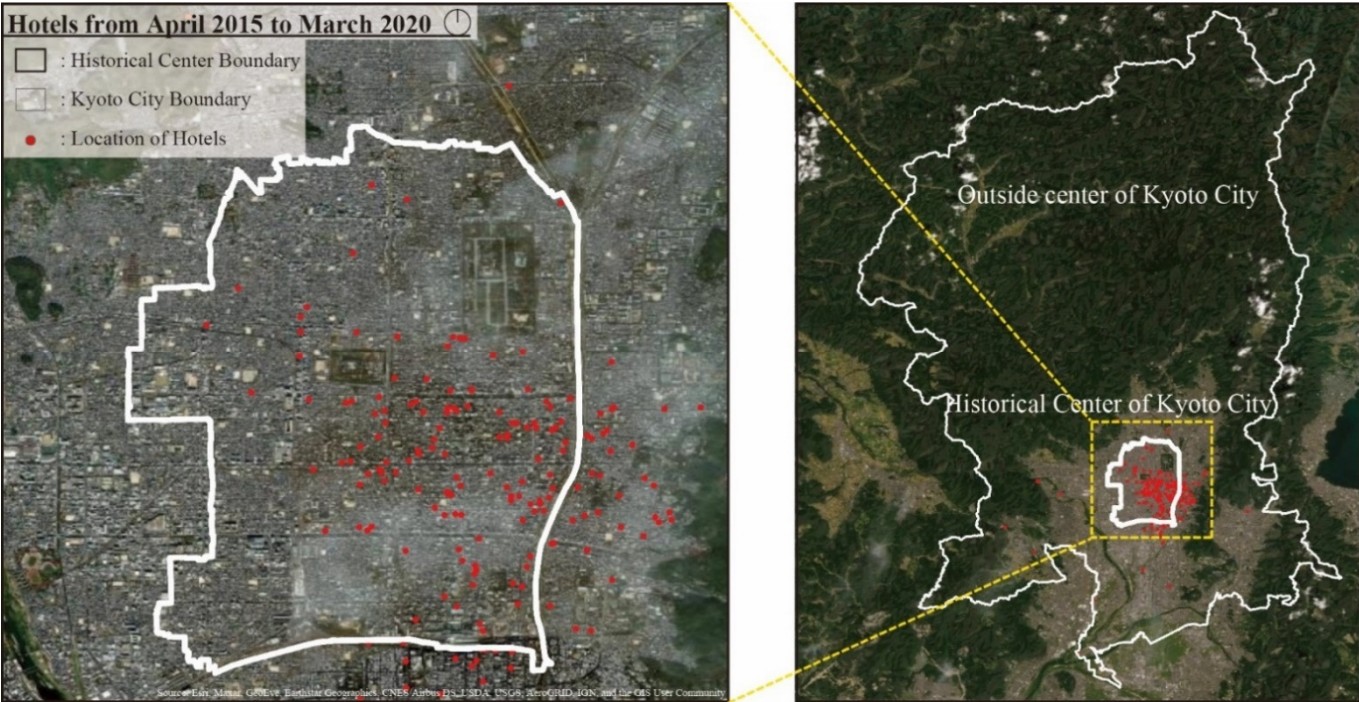

**Figure 4.** Map of hotels in Kyoto City.

Finally, the population change distribution was analyzed in Figure 5. The satellite map in Figure 5 is open source from Arc GIS PRO and complies with copyright [14]. Figure 5

shows the population change of NAs from 2015 to 2020. Figure 5 shows no clear association of NAs with increasing and decreasing populations. This signifies adjacent NAs with growing populations and NAs with declining populations, especially in the center of Kyoto City. It appears to suggest random increases and decreases in population. It indicates the validity of the geographic natural experiment that this study applied. The results suggest that the population declines were related to the location of accommodation.

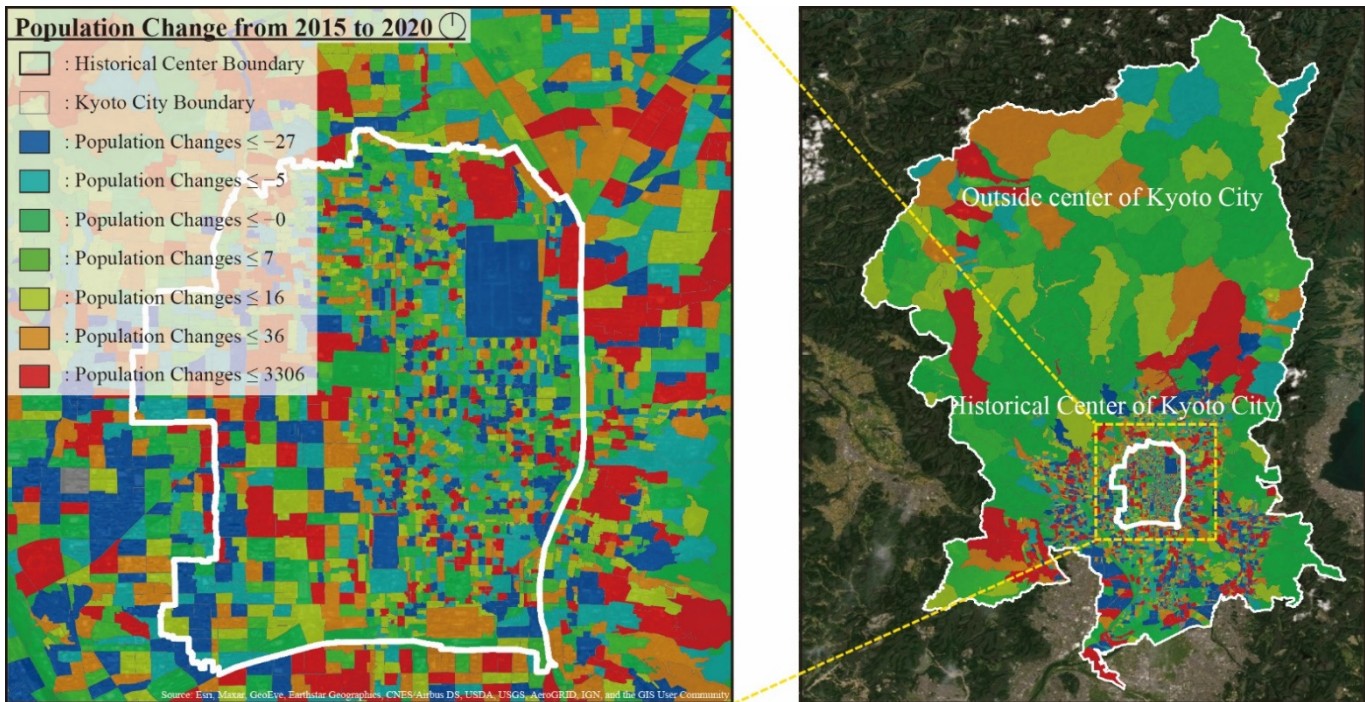

**Figure 5.** Map of population change in Kyoto City.

### 3.2. Average Population Change

In this section, we compared the population change of NAs from April 2015 to March 2020 with the existence of simple accommodation, as geographic natural experiments. Figure 6 shows the average population change in the historical center of Kyoto City. Figure 6 indicates significant differences according to the Mann–Whitney U test. In Figure 6, the error bar shows a 95% confidence interval. The left side of Figure 6 shows that the average population change of NAs with SA is −7.03, and that of NAs without SA is 1.24. This means that NAs with SA deceased the population, although NAs without SA increased the population slightly. The left side of Figure 6 shows that the NAs with SA more significantly decreased the population than NAs without SA. The results suggest that the presence of SA significantly reduced the population in NAs. This means that the population decline occurred due to tourism gentrification caused by the SA in the historical center of Kyoto City.

On the other hand, the right side of Figure 6 shows that the NAs with hotels did not decrease the population more significantly than NAs without hotels. The right side of Figure 6 shows that the average population change of NA with hotels is −11.7, and that of NA without hotels is −1.58. Figure 6 shows that the presence of hotels in NA has no statistical relationship to the population decline in the historical center of Kyoto City. These results suggest that SA might cause a population decline as part of tourism gentrification in the historical center of Kyoto City.

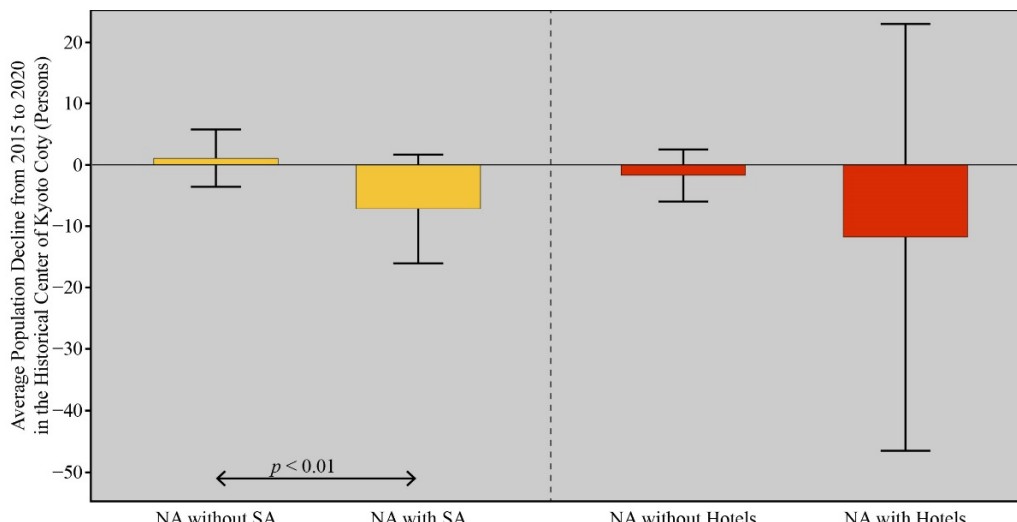

**Figure 6.** Average population change in the historical center of Kyoto City.

Figure 7 shows the average population change in the outside center of Kyoto City. Figure 7 indicates significant differences according to the Mann–Whitney U test. In Figure 7, the error bar shows a 95% confidence interval. The left side of Figure 7 shows that the average population change of NA with SA is −0.66, and that of NA without SA is 1.58. Moreover, the right side of Figure 7 shows that the average population change of NA with hotels is −3.85, and that of NA without hotels is 1.35. In the outside center of Kyoto City, compared to NA in the historical center of Kyoto City, population decline rates are lower in NA with accommodation. This means that NAs with accommodation, which are hotels and SA, deceased in population, although the NAs without accommodation increased in population. Figure 7 shows that NAs with SA significantly decreased in population, compared to NAs without SA. In addition, Figure 7 shows that the NAs with hotels significantly decreased in population, compared to NAs without hotels. The results suggest that the presence of SA and hotels significantly reduced the population of the NAs in the outside center of Kyoto City. This means that population decline might had been caused by tourism gentrification in the outside center of Kyoto City.

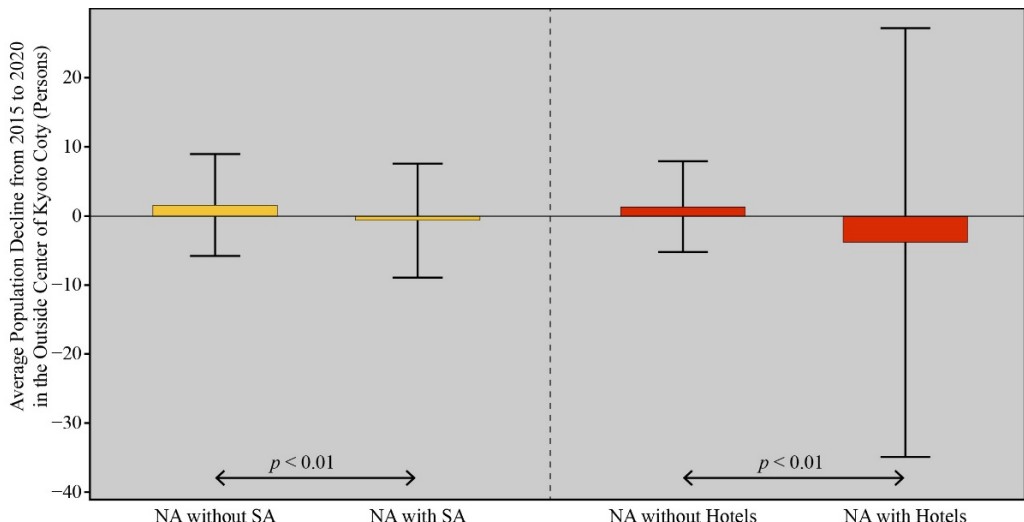

**Figure 7.** Average population decline in the outside center of Kyoto City.

## 4. Discussion

In conclusion, this study clarified the statistical relationship between the population decline and the tourism gentrification caused by SA from April 2015 to March 2020 in

the historical center of Kyoto City. The result was significant because it indicated that tourism gentrification took place as population decline occurred in the historical center of Kyoto City. In Japan, many cities have struggled with urban planning for population decline due to aging. Therefore, we cannot ignore the problem of population decline due to tourism gentrification. This was verified through residential evaluation, which analyzed the historical center of Kyoto City [25]. The result was relevant to previous studies, in which it was reported that tourism gentrification caused the displacement of long-term residents [28,31]. In addition to the findings of previous studies, this study clarified that tourism gentrification caused not only displacement, but also population decline in the case of Kyoto City. The results mean that tourism gentrification did not cause a localized and specific population decline, which Sumka et al. [9] reported as gentrification. The results strengthen the findings of Parralejo et al. [34], who analyzed the historical center of Seville. Compared to [35], this conclusion was obtained by analyzing the NAs of the entire city, including the historical center and the outside center of Kyoto City.

This study also found that the population decline was caused by the location of accommodation in the outside center of Kyoto City. As in Madrid [35], tourism gentrification in the center of Kyoto City might transmit to the outside center of Kyoto City, which is shown in Figure 3. That means that population decline is caused by the location of accommodation, not only in the historical center of Kyoto City, but also in the other center of Kyoto City. The results were different to those in the case of Valencia [36], and they suggest that tourism gentrification had a more substantial impact, as it was transmitted from the historical center to the outside center of Kyoto City. However, the reasons for the population decline are not known to be the same in the historical center and in the other center of Kyoto City. Future research needs to analyze the statistical causal relationship between whether building accommodations reduced the neighborhood's population or whether accommodations were built in neighborhoods where the population declined, depending on the historical and the other center of Kyoto City.

We need to note that SA, not hotels, caused the population decline in the historical center of Kyoto City. This study's results clarified that hotels did not significantly affect the population. Hotels provide benefits to the neighborhood's economy. In 2022, the Kyoto City Government developed a policy to encourage the construction of high-quality hotels in Kyoto City [43]. However, the rapid increase in SA caused tourism gentrification as the population declined. The reason for this population decline might be due to SA being converted from houses due to tourism gentrification. The result was significant because the government needs to develop an urban policy for SA to prevent population decline due to tourism gentrification.

## 5. Conclusions

We conclude that tourism gentrification caused the population decline in the historical center of Kyoto City. Moreover, the tourism gentrification might transmit to the outside center of Kyoto City. In terms of the declining population, tourism gentrification is a serious social problem. The significant finding of this study is the statistical relationship between the population decline and tourism gentrification. Regarding gentrification, we sometimes make comparisons between the advantages of the economic aspects and the disadvantages of the social aspects. That is because gentrification tends to provide economic benefits to neighborhoods. Of course, tourism gentrification also provides economic benefits through the growth of the tourism industry [2]. It might be possible to evaluate it in terms of the success of the urban renewal of tourist cities. However, tourism gentrification provides only accommodation, and not new middle-class settlements [8], which causes population decline. From this viewpoint, we can state that tourism gentrification has more disadvantages than advantages. In addition, for example, the concept of Airbnb is to live anywhere, and to have a once-in-a-lifetime opportunity to make the world your home [44]. This means that many travelers want to live as well as travel in tourist cities. That contradicts the claim that tourism has reduced the population in tourist cities.

Tourist cities' governments worldwide need to work in harmony with the owners of accommodation. Regarding their coexistence, this study's results suggest the need for urban policy to regulate the zoning for the locations of the total SA. Other cities have also reported urban plans that have developed zoning for accommodation facilities [45]. During the COVID-19 pandemic, accommodation stopped operating or changed their building usage to offices and houses because tourism declined rapidly. However, in the future, massive numbers of tourists may visit Kyoto again, as before the pandemic [46]. During this time, governments of tourist cities need to develop urban planning to combat tourism gentrification, based on the lessons learned before the pandemic.

This study implicates a causal relationship, in that tourism gentrification has caused population decline. However, for the analysis of the causal relationship, this study could not analyze the difference in time. Therefore, we cannot deny the possibility that population decline might cause tourism gentrification. Population decline due to tourism gentrification is an alarming issue, because population decline due to aging has received a lot of attention in Japan. In urban planning, it is not easy to analyze causal relationships using the randomized controlled trial method as a prospective study. Therefore, geographic natural experiments, which this study used, are an effective method for this study. For the analysis of the causal relationship, future research needs to analyze the urban transformation process of displacement. For the analysis, it is essential to distinguish whether the building before the accommodation was a habitation house or another type of building. That means that the tourism gentrification will be analyzed based on the building, and not on the neighborhood scale.

**Author Contributions:** Conceptualization, H.K.; methodology, H.K.; software, H.K.; validation, H.K.; formal analysis, H.K.; investigation, H.K.; resources, H.K.; data curation, H.K.; writing—original draft preparation, H.K.; writing—review and editing, H.K. and A.T.; visualization, H.K. and A.T.; supervision, H.K.; project administration, H.K.; funding acquisition, H.K. All authors have read and agreed to the published version of the manuscript.

**Funding:** This research was funded by the JSPS KAKENHI (grant number 21K14318) and the Association of Real Estate Agents of Japan (2022).

**Institutional Review Board Statement:** Not applicable.

**Informed Consent Statement:** Not applicable.

**Data Availability Statement:** The data of the accommodation list and census data are available from references as open source data [40–42]. The data presented in this study are available on request from the corresponding author, H.K.

**Conflicts of Interest:** The authors declare no conflict of interest.

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
