# Peer review of "Population Decline through Tourism Gentrification Caused by Accommodation in Kyoto City"

_sustainability, doi:10.3390/su141811736_

Round 1

Reviewer 1 Report (Previous Reviewer 2)

Thank you for the opportunity of reading and reviewing your interesting manuscript. The paper addresses an interesting and relatively novel topic and uses case study for the research part.

I noticed that my previous recommendations were partially addressed, particulary the discussion section.

I still recommend to place the limitations in the final section, i.e. Conclusions, not in the Discussion section. Moreover you must add implications of your findings to answer to the issue of relevance and importance of your research.   

Regarding the research presentation, it is a bit sketchy, and I recommend to enhance this part and present and discuss the results in more details. There are shortcomings and it is quite difficult to understand some deductions and how you came to them. You should address this issue as well.

Good luck!

Author Response

Dear Reviewer:

We appreciate the reviewer for the generous comment on the manuscript. We have attached our response letter in PDF format. We believe that the manuscript is now suitable for publication in Sustainability and look forward to hearing from you concerning your decision.

Yours sincerely

Haruka Kato

Reviewer 2 Report (Previous Reviewer 3)

Thanks for addressing my comments. 

Author Response

Dear Reviewer:

We appreciate the reviewer for the generous comment on the manuscript. We have attached our response letter in PDF format. We believe that the manuscript is now suitable for publication in Sustainability and look forward to hearing from you concerning your decision.

Yours sincerely

Haruka Kato

Reviewer 3 Report (New Reviewer)

Dear Authors,

The topic is very interesting but the paper is confused in its structure.

I recommend to clarify the objectives, methodology and results better.

Also, a theoretical framework (analyzed in depth) is needed

The description of “materials and method” should be extended, in the present version is not very clear, a special reference to the research questions is needed.

I hope these comments are helpful and will improve the manuscript.

Kind regards,

Author Response

Dear Reviewer:

We appreciate the reviewer for the generous comment on the manuscript. We have attached our response letter in PDF format. We believe that the manuscript is now suitable for publication in Sustainability and look forward to hearing from you concerning your decision.

Yours sincerely

Haruka Kato

Round 2

Reviewer 3 Report (New Reviewer)

Dear Authors, 

I appreciate most of your changes and they lead to a now complete contribution.

This manuscript is a resubmission of an earlier submission. The following is a list of the peer review reports and author responses from that submission.

Round 1

Reviewer 1 Report

The study is interesting, but needs many improvements.

Article structure. 

The article should begin with the importance of the study. Then you should emphasize the research gap, research questions, and purpose of the study. You can also add a hypothesis.

Literature review. 

How do other scientists measure and study gentrification in tourism? Please add reliable sources from leading scientific journals.

Results have shown no impact. Maybe you can add a little interview with people who live in the studied area. Do they feel the impact of tourism objects on their lives in terms of gentrification?

Discussion

What is similar and what is different from previous studies?

Please add practical implications of the results of the study.

Reviewer 2 Report

Thank you for the opportunity of reading and reviewing your interesting manuscript. The issue addressed in the article is a relatively new one in the literature, but fully relevant for the case presented in the paper. I appreciate the literature background and the soundness of the reserach as well.

The paper would benefit from a more in-depth discussion of the results. This section should be the most important in the article, and now it is quite brief. You should enhance this part and separate it from the Conclusions sections. The conclusions along with the implications both theoretical and practical, and limitations should be the final section of the paper.

Good luck! 

Reviewer 3 Report

1. there are many errors in English grammar. Polishing from a native English speaker or professional editing service is highly recommended.

2. I don't see where the authors discuss the contribution of the study and the implication of the study.

3. the paper is very descriptive. There is little to know actually analysis. 

4. what is the theory and rationale behind the results? What causes those changes? What are the channels for the changes? Any alternative explanations? 

5. the paper does not discuss how the SEM is estimated.

6. what is the economic significance of the result?

7. what is the takeaway for researchers and regulators outside Japan?